# Analysis of the 5′ Untranslated Region Length-Dependent Control of Gene Expression in Maize: A Case Study with the *ZmLAZ1* Gene Family

**DOI:** 10.3390/genes15080994

**Published:** 2024-07-29

**Authors:** Bingliang Liu, Xiaowei Liu, Min Sun, Yanxia Sun, Dayu Liu, Li Hao, Yang Tao

**Affiliations:** 1College of Food and Biological Engineering, Chengdu University, Chengdu 610106, China; sunyanxia1976@cdu.edu.cn (Y.S.); liudayu@cdu.edu.cn (D.L.); haoli@cdu.edu.cn (L.H.); 2Chengdu Academy of Agriculture and Forestry Sciences, Chengdu 611130, China; liuxw01@126.com; 3Institute for Advanced Study, Chengdu University, Chengdu 610106, China; summin1028@163.com; 4Innovative Institute of Chinese Medicine and Pharmacy, Chengdu University of Traditional Chinese Medicine, Chengdu 611137, China

**Keywords:** 5′ UTR, maize, *Lazarus 1* gene family, gene expression

## Abstract

The untranslated regions (UTRs) within plant mRNAs play crucial roles in regulating gene expression and the functionality of post-translationally modified proteins by various mechanisms. These regions are vital for plants’ ability to sense to multiple developmental and environmental stimuli. In this study, we conducted a genome-wide analysis of UTRs and UTR-containing genes in maize (*Zea mays*). Using the *ZmLAZ1* family as a case study, we demonstrated that the length of 5′ UTRs could influence gene expression levels by employing *GUS* reporter gene assays. Although maize and arabidopsis (*Arabidopsis thaliana*), as well as rice (*Oryza sativa*), have distinct functional categories of UTR-containing genes, we observed a similar lengthwise distribution of UTRs and a recurring appearance of certain gene ontology (GO) terms between maize and rice. These suggest a potentially conserved mechanism within the Poaceae species. Furthermore, the analysis of *cis*-acting elements in these 5′ UTRs of the *ZmLAZ1* gene family further supports the hypothesis that UTRs confer functional specificity to genes in a length-dependent manner. Our findings offer novel insights into the role of UTRs in maize, contributing to the broader understanding of gene expression regulation in plants.

## 1. Introduction

The untranslated regions (UTRs) in eukaryotic organisms are crucial for the regulation of gene expression, despite not encoding proteins themselves. They significantly influence gene expression through interactions with various molecular elements, such as ribosomes and initiation factors. These interactions can alter translation efficiency and protein synthesis rates [1,2,3]. In plants, both the 5′ UTR and 3′ UTR are instrumental in modulating gene expression across diverse tissues, developmental stages, and in response to environmental stresses. This regulatory flexibility enables plants to adapt to a broad spectrum of growth conditions [4,5,6]. For example, examining the function of the 3′ UTR in rice (*O. sativa*) contributes to our understanding of the molecular mechanisms that confer disease resistance. Investigating the variations in UTRs among different disease-resistant varieties can elucidate their contribution to resistance, thereby providing valuable insights for the development of crops with enhanced disease resistance [7]. Beyond their involvement in stress responses, 5′ UTRs in plants also play a significant role in development, influencing processes such as pollen formation and embryonic development [8].

Sequences within the 3′ UTR serve as critical binding sites for microRNAs, which regulate gene expression through interacting with microRNA. An instance is observed in the Voltage-Dependent Anion Channel (VDAC) family member, VDAC3, in arabidopsis (*A. thaliana*), which produces two mRNA variants characterized by either short or long 3′ UTRs. The distal 140 nt sequence in the long UTR plays a pivotal role in the co-translational import of VDAC3 and other unrelated mRNAs to the mitochondrial surface [9]. The *FLOWERING LOCUS C* (*FLC*) serves as a crucial gene regulating arabidopsis flowering time. Its 3′ UTR region contains multiple binding sites for microRNAs, which interact with the 3′ UTR of *FLC* to affect the stability of its mRNA, thereby regulating the flowering time. This regulatory mechanism enables plants to adjust their lifecycle in response to environmental conditions [10]. The *OsSERK2* gene in rice plays a role in resistance to multiple diseases. According to research, specific sequences within its 3′ UTR are able to respond to pathogen infection by interacting with particular microRNAs or RNA-binding proteins, regulating the expression level of *OsSERK2*, and thereby influencing the disease resistance of rice [11]. *Zm401* is associated with drought tolerance in maize. Its 3′ UTR contains conserved *cis*-acting elements that can affect the stability and translation efficiency of *Zm401* mRNA. Under drought conditions, the activity of these *cis*-acting elements changes, thereby adjusting the expression level of *Zm401* and helping maize adapt to the drought environment [12].

The stability, transport, and translation efficiency of mRNA are regulated by elements within the 5′ UTR, which in turn affect the functionality and subcellular localization of the resulting proteins [13]. For instance, *AtNRAMP3* is a gene responsible for metal ion transport in arabidopsis. Specific sequences within its 5′ UTR can influence the translation initiation rate of its mRNA, thereby regulating the expression of the AtNRAMP3 protein. This regulation is crucial for maintaining intracellular metal ion balance [14]. In wheat, the 5′ UTR of *TaNAC29* contains multiple *cis*-acting elements that can regulate the expression of the TaNAC29 protein, helping wheat adapt to adverse environmental conditions [15]. In potatoes, editing the 5′ UTR of the *VInv* gene can achieve stable cold-induced sweetening resistance [16]. Consequently, a detailed examination of the functional mechanisms of plant UTRs can significantly advance our comprehension of plant biological characteristics and functions.

The length of the 5′ UTR in plants significantly influences the functional specificity of genes [17,18]. In arabidopsis, some genes involved in the stress response have longer 5′ UTRs. These UTR regions may contain *cis*-acting elements that respond to stress signals [6]. In addition, genes involved in metabolic processes tend to have longer 5′ UTRs, whereas those associated with signal transduction often feature shorter ones [19]. Previous research has detailed the 5′ UTR characteristics of model plants such as arabidopsis and rice [20]. The 5′ UTRs ranging from 1 to 500 bp are linked with responses to environmental stresses like salt, cadmium ion, and cold. UTRs spanning 501–1000 bp correlate with cellular protein modification processes, biosynthetic activities, and signal transduction mechanisms. Those measuring 1001–2000 bp are associated with the regulation of ion transmembrane transportation, phospholipid catabolic processes, and leaf formation, among other functions. UTRs exceeding 2000 bp are related to processes such as sister chromatid cohesion, cytokinin-activated signaling pathways, and meiotic chromosome segregation. However, the underlying mechanisms that link UTR length with gene expression modulation remain to be elucidated. Furthermore, it is uncertain whether insights regarding UTRs derived from arabidopsis and rice are applicable to other crop species such as maize (*Z. mays*), highlighting an area for further investigation.

Lazarus 1 (LAZ1) is a conserved transmembrane protein that exhibits sequence homology and structural similarity to members of the DUF300 family [21]. Despite their shared characteristics, the physical and chemical properties, secondary structures, transmembrane structures, transport substrates, subcellular localization, and expression regulation of coding genes within this family are highly diverse. In arabidopsis, two LAZ1 proteins have been identified as localized on the vacuolar membrane, where they play crucial roles in maintaining vacuole integrity and mediating brassinosteroid (BR) signaling pathways [22]. Our previous research successfully cloned eight members of the *ZmLAZ1* gene family in maize [23,24]. The differential expression of these genes across various organs, developmental stages, and under abiotic stress conditions implies functional diversity within the family. Furthermore, we identified that the ZmLAZ1-4 protein may function as a zinc transporter on plasma and vacuolar membranes [25]. Notably, the 5′ UTR sequence lengths of these *ZmLAZ1* members display a gradient distribution, which presents a potential avenue for investigating the relationship between 5′ UTR length and gene function.

In this study, we conducted a genome-wide analysis of the 5′ and 3′ UTRs in maize. Our findings suggest the potential conservation of relative proportions and functional specificity across these regions. Using the *ZmLAZ1* family as a case study, we employed GUS activity assays and *cis*-acting element predictions to explore the potential relationship between 5′ UTR length and gene function. Our data lend support to the hypothesis that UTRs contribute to gene regulation and confer functional specificity to genes in a length-dependent manner. These insights open a new avenue for further investigation into the functional roles of plant genes, emphasizing the significance of UTRs in gene expression and functionality.

## 2. Materials and Methods

### 2.1. Genome-Wide Identification of the UTRs from Maize, Rice, and Arabidopsis

The bioinformatics analysis methods were mainly referenced from Srivastava, Lu, Zinta, Lang, and Zhu [20]. Briefly, the Exon, 5′ UTR, and 3′ UTR lengths of each gene transcript were calculated using Python scripts based on the gtf annotation information for the maize genome (B73, version 4; Jiao et al. [26]). For genes with more than one transcript, the longest transcript was screened for final analysis. The length of the UTR was then divided into four gradients (i.e., 1–500 bp, 501–1000 bp, 1001–2000 bp, and >2000 bp), and statistically analyzed using the “awk” command of the bash script. For cross-species homolog analysis, blastn (2.7.1) was used to compare all genes of maize with those of arabidopsis (TAIR10) and rice (IRGSP-1.0), respectively. The unique gene with the highest score was selected as the target gene for subsequent analysis.

### 2.2. Gene Ontology (GO) Enrichment Analysis on UTR-Containing Genes

The 5′ UTR and 3′ UTR of the maize genome were enriched by GO analysis according to four gradients of sequence length. The gene database (http://www.geneontology.org/, accessed on 12 December 2023) was used to map each term and calculate the number of genes. Then, hypergeometric tests were applied to identify significantly enriched GO terms. The data were finally visualized using Graph Pad Prism (9.5.0).

### 2.3. Sample Preparation

The seeds of the maize inbred line B73 were surface sterilized with 30% H_2_O_2_, germinated in a petri dish, and transplanted into a plastic mesh grid for hydroponic culturing at 28 °C under a photoperiod of 14 h light/10 h dark. At the three-leaf stage, the leaf and root were sampled, ground in liquid nitrogen, and used for total RNA extraction using an RNAiso plus kit (TaKaRa, Kyoto, Japan) with technical replicates, respectively. After removing probable genomic DNA using RNase-free DNase (TaKaRa, Japan), the RNA samples were quantified on NanoDrop 2000 (ThermoScientific, Waltham, MA, USA), and reverse transcribed into cDNA using a PrimeScript^TM^ reagent kit (TaKaRa, Kyoto, Japan).

### 2.4. Gene 5′ UTR Cloning

According to the 5′ UTR sequences of *ZmLAZ1s* (Appendix A), specific primers were designed by the Premier 5.0 software (http://www.premierbiosoft.com/, accessed on 9 May 2022) and were used for amplification of the 5′ UTR of the *ZmLAZ1* genes. The PCR reaction system was 25 μL in total volume and contained 1 μL cDNA sample from the blank control (0 h), 0.3 μL Phanta Max Super-Fidelity DNA Polymerase (Vazyme Biotech, Nanjing, China) (2.5 U/μL), 5 μL 5 × bufer, 2 μL dNTP mix (0.4 mM), and 0.5 μL each primer (10 pM), 15.7 μL H_2_O. The temperature cycle was as follows: pre-denaturation at 95 °C for 3 min; 35 cycles of denaturation at 95 °C for 10–15 s; annealing at the annealing temperature of each pair of primers for 15 s; extension at 72 °C for 1.5 min; and re-extension at 72 °C for 5 min. The PCR products were separated by 1.5% agarose gel electrophoresis. The target fragments were recovered using DNA recovery and a purification kit (TianGen, Beijing, China), subcloned into vector pCAMBIA1381-*GUS* (Takara, Kyoto, Japan) to generate transient expression vectors pCAMBIA1381-*ZmLAZ1-5′ UTR-GUS*, and sequenced at Sangon Biotech (Shanghai, China). All the primers used in this study are listed in Appendix A.

### 2.5. GUS Histochemical Staining

The *GUS* report vectors were transformed into *Agrobacterium tumefaciens* strain GV3101, which was infiltrated into the abaxial air space leaves of 3-week-old tobacco (*Nicotiana benthamiana*). The pCAMBIA1381-*GUS* was used as a control. After infiltration, the tobacco was cultured 48 h at 25 °C under 14 light/10 dark. Injected tobacco leaves were cut into 2 cm × 2 cm used for Gusblue kit (Huayueyang, Beijing, China). After decolorization with 70% ethanol for 3–5 times until the background shows negative control (i.e., appears white), observe with the naked eye and record the results by taking photos.

### 2.6. GUS Activity Measurement

The GUS enzyme activity determination method mainly follows Jefferson et al. [27]. Around 0.1 g of plant tissue was transferred into a 1.5 mL centrifuge tube pre-cooled with liquid nitrogen and 200 µL of the extraction solution was added. It was allowed to thaw at room temperature and was mixed well. It was centrifuged at 4 °C and 12,000 r/min for 15 min. A total of 50 µL of the GUS crude protein extract was taken out and added to 950 µL of ddH_2_O, then 5 mL of G250 solution was added and mixed well. After placing it at room temperature for 10 min, the absorbance value was measured at 595 nm based on the standard curve to calculate the protein concentration. Additionally, 70 µL of the GUS crude protein extract was taken out and mixed with 80 µL of the GUS reaction solution. A total of 30 µL of the mixture was immediately pippeted out and placed into a 60 µL mixture of 0.2 mol/L Na_2_CO_3_ to terminate the reaction. The remaining reaction solution in a water bath was incubated at 37 °C for 1 h, and then 240 µL of 0.2 mol/L Na_2_CO_3_ was added to terminate the reaction. This was mixed well and 90 µL of the mixture was taken out for reading in a 96-well microplate reader. A microplate luminometer was used to read the values at a excitation wavelength of 340 nm and emission wavelength of 465 nm for both 0 h and 1 h time points. The MU concentration was calculated at 0 h and 1 h based on the standard curve, and the GUS activity value was the concentration difference between the MU concentration at 1 h and that at 0 h divided by the incubation time (1 h) and protein concentration, as a reference for GUS activity.

### 2.7. Analysis of Cis-Acting Elements

The 5′ UTR sequences of the *ZmLAZ1* genes were retrieved from the MaizeGDB database (http://www.maizegdb.org/, accessed on 10 January 2024) and used for the prediction of *cis*-acting elements by the online tool PlantCARE (http://bioinformatics.psb.ugent.be/webtools/plantcare/html/, accessed on 12 January 2024). Furthermore, the TBtools software (version 2.029) was employed to visualize the composition of *cis*-elements in the 5′ UTR [28].

### 2.8. Statistical Analysis

All experiments were performed with three replicates. The data were shown as the mean ± standard deviation and analyzed using Student′ s *t*-test at * *p* < 0.05 and ** *p* < 0.01 levels.

## 3. Results and Discussion

### 3.1. Whole-Genome Survey of UTRs in Maize

To explore the potential role of UTRs in maize, we conducted a comprehensive whole-genome survey of UTRs in maize. The results showed that UTRs constitute approximately 24.7% of the transcribed sequences, suggesting their significance in the maize genome (Figure 1A). When compared with previous studies on arabidopsis and rice [20], maize UTRs were found to represent a larger fraction of the transcribed sequences than those in rice and arabidopsis. Given the maize genome’s substantial size of about 2.3 Gb—significantly larger than the rice genome (~380 Mb) and the arabidopsis genome (~140 Mb)—this discrepancy in the UTR/CDS ratio may reflect differences in genome size among these species. Notably, in all three plants, the proportion of 3′ UTRs consistently exceeded that of 5′ UTRs.

Further classification of maize UTRs into four length categories revealed a distribution pattern in the 1−500 bp, 501−1000 bp, and 1001−2000 bp ranges for both 5′ and 3′ UTRs: 87.64%, 9.87%, and 2.30% and 78.46%, 17.03%, and 4.04%, respectively (Figure 1B). This pattern is similar to that observed in rice (87.7%, 8.4%, and 3% and 73%, 19%, and 6%) [20]. However, for sequences longer than 2000 bp, maize exhibited unique proportions of its 5′ UTR (0.19%) and 3′ UTR (0.46%) compared to rice and arabidopsis. Moreover, the average sequence length of maize’s 5′ UTR was 430 bp, longer than that in rice (259 bp) and arabidopsis (155 bp), while the mean 3′ UTR length in maize (382 bp) fell between that of rice (469 bp) and arabidopsis (242 bp).

By analyzing protein sequences across species, we identified 31,380 maize genes with homologs in both rice and arabidopsis (Figure 1B). The average sequence lengths of these maize genes’ 5′ UTR (273 bp) and 3′ UTR (399 bp) were also longer than their counterparts in rice and arabidopsis [29]. In orthologs, the UTR distributions of maize and rice exhibited similar length gradients, likely due to their shared Poaceae family membership and closer genetic relationship. Collectively, these findings suggest that the relative proportion of UTR length is also conserved in maize.

### 3.2. Functional Enrichment Analysis of UTR-Containing Genes in Maize

Utilizing methodologies previously applied to arabidopsis and rice [20], we categorized UTR-containing genes within the maize genome based on the lengths of their 5′ UTR and 3′ UTR and conducted a gene ontology (GO) enrichment analysis. Our analysis focused on the top five enrichment terms within the biological process category. Consistent with expectations, genes with short (1−500 bp), medium (501−1000 and 1001−2000 bp), and long (>2000 bp) UTRs in maize were each enriched in distinct functional classes (Figure 2A). For genes with 5′ UTRs measuring 501−1000 bp, the most enrichment term was “Phosphorus metabolic process”, followed by “Protein phosphorylation”, “Cellular protein modification process”, “Cellular communication”, and “Regulation of cellular processes”. In contrast, genes with 1−500 bp 5′ UTRs predominantly showed enrichment in “Developmental process”, “Response to abiotic stimulus”, “Organic substance transport”, “Anatomical structure development”, and “mRNA transport”. The enrichment index for genes with 5′ UTRs exceeding 2000 bp was comparatively lower, likely due to their smaller proportion. Intriguingly, the terms “Iron ion transport” were enriched across both the 1001−2000 and >2000 bp length categories, indicating specific biological processes that may be influenced by UTR length.

While most enrichment terms for 3′ UTR-containing genes in maize differ from those associated with 5′ UTRs, common terms such as “Anatomical structure development” and “RNA processing” are observed in both categories (Figure 2B). Similar to 5′ UTRs, genes with 3′ UTRs in the 501−1000 bp range exhibit a higher enrichment index, with “RNA metabolic process” displaying the highest index within this category. In the category representing the largest proportion of UTRs, the 1−500 bp range, the top five enrichment terms included “Response to oxidative stress”, “Anatomical structure development”, “Reactive oxygen species metabolic process”, “Hydrogen peroxide metabolic process”, and “Detoxification”. Conversely, in the categories representing the smallest proportions (>2000 bp), the enrichment terms were “Regulation of SREBP signaling pathway”, “RNA processing”, “Cellular response to sterol depletion”, “Positive regulation of cell communication”, and “Regulation of signal transduction”. This diversity in enrichment terms across different UTR lengths supports the complexity and specificity of regulatory mechanisms mediated by UTRs in gene expression.

Previous research has revealed variability in functional enrichment associated with UTR lengths in arabidopsis and rice [20]. Intriguingly, our analysis revealed that the term “transport” was consistently enriched among genes with 5′ UTRs across all three species (arabidopsis, rice, and maize) (Figure 2A). This finding suggests a potentially conserved role of 5′ UTRs in material transport across these species. Additionally, maize and rice shared several similar enrichment terms, including “Cellular protein modification process” and “Ion transport” for genes with 5′ UTRs, as well as “Metabolic processes”, “Nucleobase-containing compound metabolic process”, and “Regulation of signal transduction” for genes with 3′ UTRs (Figure 2) [20]. Given the similarity in the lengthwise distribution of UTRs between maize and rice (Figure 1), it is plausible that UTRs in Poaceae species exhibit conserved functional mechanisms. These findings lend further support to the hypothesis that UTRs contribute to gene regulation and confer functional specificity to genes in a length-dependent manner.

### 3.3. Characterization of the 5′ UTR Region of the ZmLAZ1 Gene Family

The regulatory role of the 5′ UTR in gene expression in eukaryotes is well recognized. Nonetheless, the detailed mechanisms linking 5′ UTR length to gene expression remain poorly understood [20,29]. Our prior research characterized the *ZmLAZ1* gene family in maize, uncovering not only functional specificity but also considerable variation in the length of their 5′ UTRs [23,25]. Therefore, we selected the *ZmLAZ1* gene family for further detailed analysis. We cloned the 5′ UTRs of the *ZmLAZ1* gene family and found that *ZmLAZ1-3* lacks the 5′ UTR component, while *ZmLAZ1-6*, suspected to be a pseudogene, was excluded from our analysis. We successfully amplified the 5′ UTRs of the remaining seven members (Figure 3; Appendix A). Sequence alignment confirmed that these amplified fragments matched perfectly with reference sequences from the MaizeGDB database (http://www.maizegdb.org/, accessed on 5 May 2022). The lengths of the 5′ UTRs in the *ZmLAZ1* gene family varied from 149 bp to 1391 bp. Specifically, *ZmLAZ1-1* and *ZmLAZ1-2* possessed the longest 5′ UTRs, measuring 1391 bp and 1237 bp, respectively. The medium-length 5′ UTRs of *ZmLAZ1-7*, *ZmLAZ1-8*, and *ZmLAZ1-9* measured 493 bp, 376 bp, and 355 bp, respectively, while *ZmLAZ1-4* and *ZmLAZ1-5* had the shortest 5′ UTRs, at 162 bp and 149 bp, respectively. These variations in 5′ UTR length and sequence among the family members suggest potential impacts on the expression and function of these genes.

### 3.4. Effect of 5′ UTR Length on Gene Expression

To elucidate the correlation between 5′ UTR length and gene expression, we conducted GUS activity assays (Figure 4). Tobacco leaves were inoculated with bacteria containing a blank vector as a control, alongside experimental samples inoculated with other bacteria containing the reporters. The results demonstrated higher GUS expression levels in leaves harboring the 5′ UTRs of *ZmLAZ1-1* and *ZmLAZ1-2* (Figure 4b). Leaves with the 5′ UTRs of *ZmLAZ1-4* and *ZmLAZ1-7* exhibited medium staining intensity, whereas those with the 5′ UTRs of *ZmLAZ1-5*, *ZmLAZ1-8*, and *ZmLAZ1-9* showed minimal coloration. Quantitative analysis of GUS activity corroborated the staining observations (Figure 4c). Enzyme activities for *ZmLAZ1-1* 5′ UTR- and *ZmLAZ1-2* 5′ UTR-derived reporters were significantly elevated, approximately fivefold higher than the control. *ZmLAZ1-4* 5′ UTR and *ZmLAZ1-7* 5′ UTR-derived reporters also displayed increased enzyme activity, about two-fold higher than the control. The reporter of *ZmLAZ1-9* 5′ UTR’s enzyme activity was notably higher than the control, while *ZmLAZ1-5* 5′ UTR- and *ZmLAZ1-8* 5′ UTR-derived reporters showed no significant difference in GUS activity compared to the control. These findings substantiate the hypothesis that the length of the 5′ UTR enhances gene expression.

To further explore the expression patterns of the seven *ZmLAZ1* gene family members across different maize varieties and tissues, we utilized the Plant Public RNA-seq Database (PPRD), which houses over 19,664 maize RNA-seq libraries [30]. This extensive collection allows for a comprehensive analysis of gene expression levels using big data. Our analysis revealed a wide range of FPKM values for these *ZmLAZ1* members across various tissues and developmental stages in maize, indicating diverse expression profiles (Appendix A). We then aggregated the total expression abundance of all FPKM values across different libraries (Figure 5). Notably, *ZmLAZ1-1*, which possesses the longest 5′ UTR sequence, exhibited the highest expression abundance (Figure 4b, Figure 5 and Appendix A). However, *ZmLAZ1-2*, with the second-longest 5′ UTR sequence, demonstrated lower expression abundance. *ZmLAZ1-9*, which has the shortest 5′ UTR sequence, did not exhibit the lowest expression levels. This observation suggests the complex regulatory mechanisms governing eukaryotic gene expression, where the 5′ UTR represents just one of many contributing factors. Collectively, we can infer that the length of the 5′ UTR does, to some extent, affect the expression of *ZmLAZ1* family members in maize.

### 3.5. Cis-Elements Analysis of 5′ UTR of ZmLAZ1 Gene Family

The *cis*-elements within the 5′ UTR are crucial sequence features that significantly influence gene expression [5,20]. To investigate the potential transcriptional regulatory mechanisms mediated by the 5′ UTR in the *ZmLAZ1* gene family, we performed *cis*-acting element predictions on the 5′ UTRs of seven *ZmLAZ1* members. As expected, the analysis revealed that genes with longer 5′ UTRs harbored a greater number of *cis*-elements, with the majority of *ZmLAZ1* gene 5′ UTRs containing stress response elements (STREs) (Figure 6; Appendix A). We categorized the potential functions of these *cis*-acting elements, finding that those on the 5′ UTRs of *ZmLAZ1-1* and *ZmLAZ1-2* were associated with “Abiotic and biological stresses”, “Phytohormone response”, and “Plant growth and development” (Figure 6; Appendix A). The 5′ UTRs of *ZmLAZ1-4* and *ZmLAZ1-5*, which have the fewest *cis*-acting elements, were linked to “Abiotic and biological stresses”. Meanwhile, the 5′ UTRs of *ZmLAZ1-7*, *ZmLAZ1-8*, and *ZmLAZ1-9*, which possess moderate 5′ UTR lengths, also exhibited a moderate number and diversity of *cis*-acting elements.

Utilizing the PPRD database, we assessed the response of these genes to various treatments and correlated these responses with the *cis*-acting elements identified in their 5′ UTRs (Figure 6; Appendix A). Terms related to light response were predicted in the *cis*-acting elements of the 5′ UTRs of *ZmLAZ1-1*, *ZmLAZ1-2*, *ZmLAZ1-4*, *ZmLAZ1-7*, and *ZmLAZ1-8* (Figure 6; Appendix A). Correspondingly, all but *ZmLAZ1-7* among these genes exhibited light response-related differential expression in the PPRD database (Appendix A). The 5′ UTRs of *ZmLAZ1-7* and *ZmLAZ1-9* contained elements associated with low temperature, aligning with their differential expression under cold treatment (Appendix A). In the realm of “Phytohormone response”, *ZmLAZ1-7*, which features MeJA-related elements in its 5′ UTR, showed differential expression following JA treatment according to the database (Figure 6; Appendix A). Notably, *ZmLAZ1-4* and *ZmLAZ1-5*, lacking “Phytohormone response”-associated elements in their 5′ UTRs, also did not exhibit differential expression in relation to phytohormone treatments (Figure 6; Appendix A). Thus, the results of the transcriptome big data positively support the findings from the analysis of 5′ UTR *cis*-acting elements.

Subsequently, we extended our analysis to the 5′ UTR *cis*-acting elements of the arabidopsis and rice *LAZ1* gene families, aligning these 5′ UTRs with the phylogenetic tree of LAZ1 across the three species (Appendix A). Although there appears to be no direct correlation between 5′ UTR length and the phylogenetic clustering of LAZ1 amino acids, we found an increased number of *cis*-acting elements in genes with longer 5′ UTRs, such as *OsLAZ1-2*, *OsLAZ1-6*, *OsLAZ1-7*, *AtLAZ1-4*, *AtLAZ1-5*, and *AtLAZ1-7*. This finding supports the idea that the length of the 5′ UTR contributes to its functional diversity, at least to a certain extent. Additionally, we identified some conserve *cis*-acting elements across all three species, including those associated with abscisic acid responsiveness, anoxic specific inducibility, light responsiveness, low-temperature responsiveness, and the MYB binding site involved in drought inducibility (Appendix A; Appendix A). This information may imply the conserved functions of *LAZ1* members in plant responses to various environmental stresses and provide potential guidance for further experiments to study the specific biological functions of *LAZ1* members.

### 3.6. The Potential Relationship between 5′ UTR Length and Gene Function in the LAZ1 Gene Family

The LAZ1 family, characterized by its transmembrane proteins with six DUF300 domains, is posited to play a crucial role in transmembrane transport across eukaryotes [22,31,32]. Our previous work identified the *ZmLAZ1* gene family in maize, revealing that *ZmLAZ1-4* encodes a zinc transporter that modulates zinc homeostasis under the negative regulation of the brassinosteroid (BR) signaling transcription factor ZmBES1/BZR1-11 [23,25]. Additionally, we found that the *ZmLAZ1-3* gene is a negative regulator of drought tolerance, and can be used to improve maize drought tolerance [24]. However, as with many studies on plant transmembrane transporters, our understanding of LAZ1’s functions remains hampered by challenges in expression purification systems and subcellular localization techniques. In this study, we explored the 5′ UTR of the *ZmLAZ1* family, revealing a gradient distribution in 5′ UTR length (Figure 3). Results from *GUS* reporter assays suggest a positive correlation between 5′ UTR length and gene expression (Figure 4). Further, our analysis of *cis*-acting elements indicates that the distribution of 5′ UTR functions may also correlate with sequence length (Figure 6 and Appendix A). In light of existing hypotheses regarding UTR function in plants [20], it is plausible that the 5′ UTR in the *ZmLAZ1* gene family not only modulates gene expression but also imparts functional specificity to genes in a length-dependent manner.

Our earlier study observed a notable increase in *ZmLAZ1-7* expression following NaCl treatment, suggesting its potential involvement in salt stress resistance [23]. Correspondingly, *cis*-acting elements associated with “defense and stress responsiveness” were identified within the 5′ UTR region of *ZmLAZ1-7* (Appendix A). Despite the lack of correlation between the presence of the 5′ UTR in *LAZ1* genes from maize, rice, and arabidopsis with amino acid sequence homology, the presence of over 50 light responsiveness-related terms (Appendix A) suggests a potential conserved role for the *LAZ1* family in regulating plant light response. Moreover, a recent study has found that rare variations of upstream open reading frames (uORFs) in maize 5′ UTRs affect the abundance of the main ORF proteins, thereby affecting plant metabolism and phenotypic changes [33]. This indicates that using the 5′ UTR to design or regulate changes in proteins may help address problems in synthetic biology, genome editing, crop improvement, and human disease. Therefore, the analysis of 5′ UTRs not only provides a new perspective for the experimental design and analysis of the *LAZ1* and other gene families but also serves as a valuable reference for the functional study of other gene families.

## 4. Conclusions

In this study, we employed bioinformatics methods to analyze UTR-containing genes across the entire maize genome. Our analysis revealed that maize UTRs constitute a slightly larger proportion of the transcribed sequence compared to those in rice and arabidopsis, with a conserved gradient in their length distribution. GO enrichment analysis highlighted the relationship between UTR length distribution and specific functions in maize. Notably, we observed recurrent GO terms associated with material transport and signal transduction in both maize and rice, likely reflecting their shared genetic heritage within the Poaceae family. Using the *ZmLAZ1* gene family as a case study, we conducted *GUS* reporter gene assays and *cis*-acting element analyses on the 5′ UTR. Our results indicate that the 5′ UTR of *ZmLAZ1* can modulate gene expression and likely confer functional specificity in a length-dependent manner. These insights enhance our understanding of the potential roles of maize 5′ UTRs and offer a novel perspective for unraveling the intricate gene regulatory mechanisms in plants.

## Figures and Tables

**Figure 1 genes-15-00994-f001:**
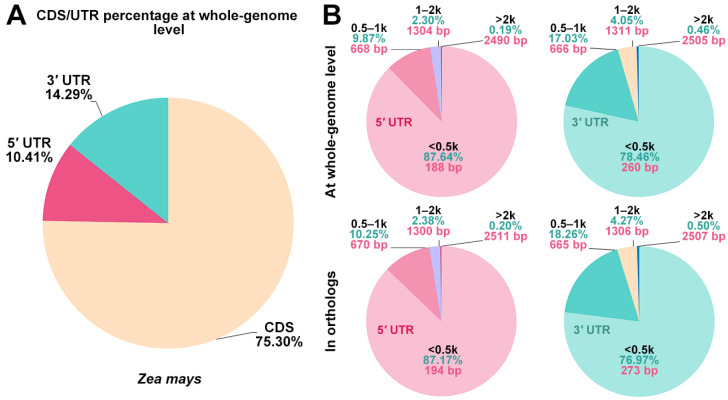
The percentages of coding sequence (CDS) and untranslated region (UTR) elements in maize and their lengths. (**A**) A genome-scale survey showing the percentages of transcript bases involved in protein coding (CDS) and in gene regulation (UTR) in maize (*Z*. *mays*). (**B**) Percentages of 5′ and 3′ UTRs in different length categories along with the average UTR length (bp) at the whole-genome level and in orthologs.

**Figure 2 genes-15-00994-f002:**
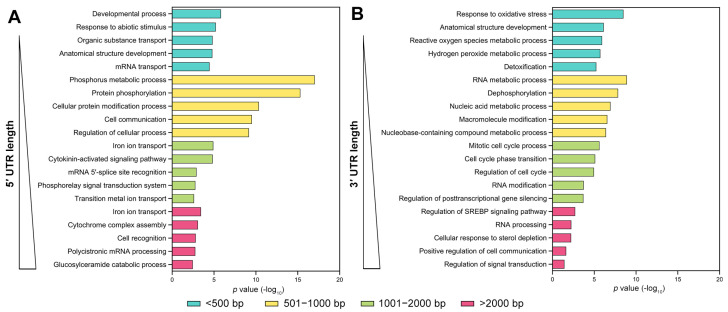
Functional enrichment analysis of untranslated region (UTR)-containing genes in maize. UTR-containing genes of maize genomes were sorted according to their 5′ UTR (**A**) and 3′ UTR (**B**) lengths before functional enrichment analysis was independently performed for genes with short (1−500 bp), medium (501−1000 and 1001−2000 bp), and long (>2000 bp) UTRs. On the basis of *p* values, the top five GO terms related to biological processes are represented independently for 5′ and 3′ UTRs.

**Figure 3 genes-15-00994-f003:**
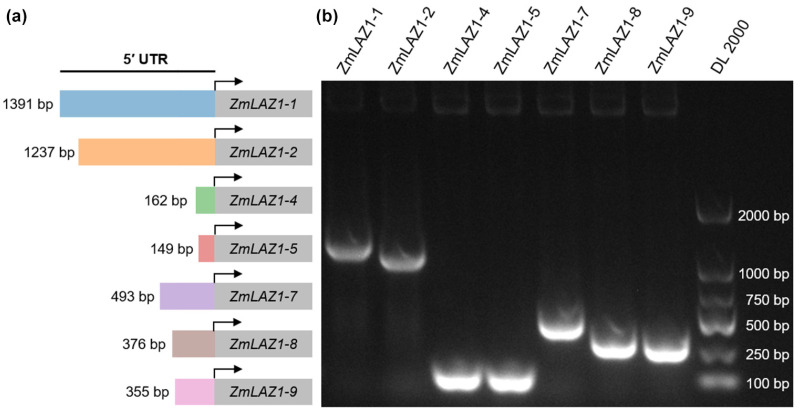
Percentages of coding sequence (CDS) and untranslated region (UTR) elements in maize (**a**) Schematic representation of the *ZmLAZ1* gene family members with 5′ UTRs. (**b**) Specific fragments of the 5′ UTR sequences of the *ZmLAZ1* genes cloned from the maize cDNA sample and separated by agarose gel electrophoresis.

**Figure 4 genes-15-00994-f004:**
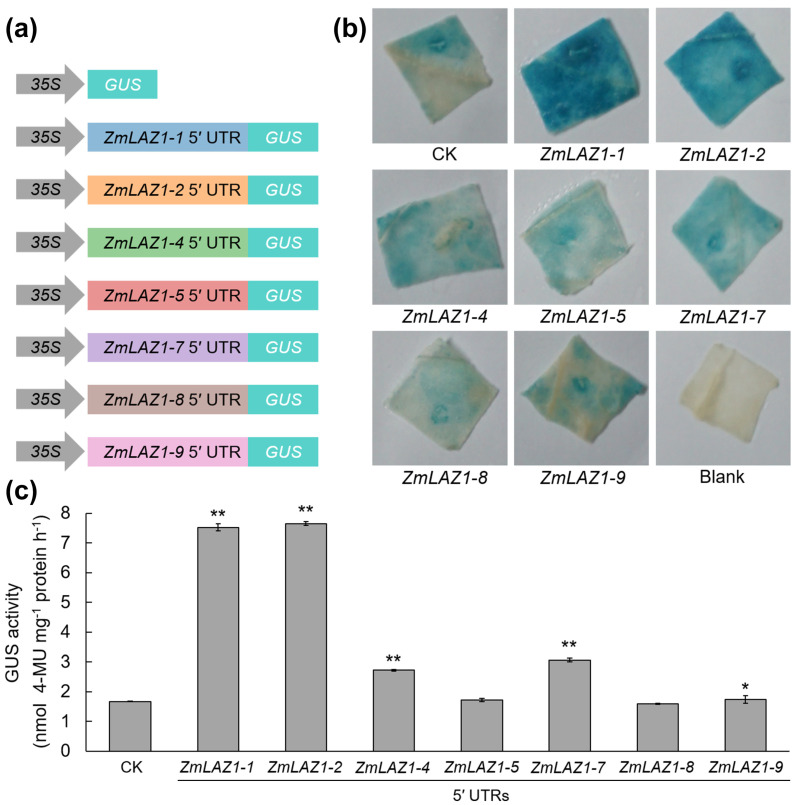
GUS reporter assays. (**a**) Schematic diagrams of the reporters used for the GUS assay. (**b**) Histochemical GUS staining in tobacco leaves. The depth of color represents the different expression levels of the GUS protein. (**c**) GUS enzyme activity assay in tobacco leaves. Student’s *t*-test; **, *p* < 0.01; *, *p* < 0.05.

**Figure 5 genes-15-00994-f005:**
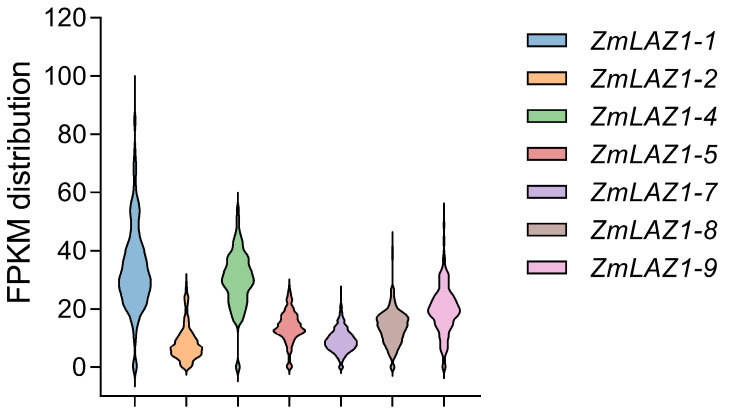
FPKM distribution of the *ZmLAZ1* family members in all libraries of the PPRD database.

**Figure 6 genes-15-00994-f006:**
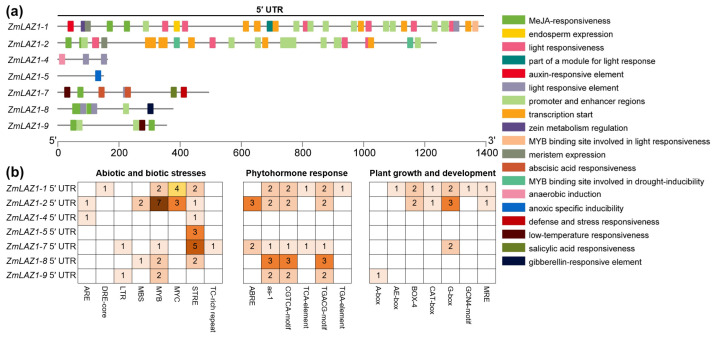
Analysis of *cis*-acting elements in the 5′ UTR of the *ZmLAZ1* gene family. (**a**) The *cis*-acting elements of the 5′ UTR region, and different color blocks represent different elements. (**b**) The putative *cis*-elements were quantified and functionally classified based on their established roles in gene transcriptional regulation.

## Data Availability

The datasets supporting the conclusions of this article are included within the article (and its additional files).

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
