# Peer review of "Analysis of the 5′ Untranslated Region Length-Dependent Control of Gene Expression in Maize: A Case Study with the ZmLAZ1 Gene Family"

_genes, 2024, doi:10.3390/genes15080994_

Round 1

Reviewer 1 Report

Comments and Suggestions for Authors

The brief report titled “Analysis of 5′ UTR Length-Dependent Control of Gene Expression in Maize: A Case Study with the ZmLAZ1 Gene Family,” analyzed and written by Liu et al is a summary of genome-wide analysis of the 5′ and 3′ UTRs in maize.

The authors have carried out a detailed analysis and have generated a great resource that other researchers can look into for analysis of certain genes and planning further experiments.

UTRs are significant in monitoring gene expression through various mechanisms. The 5′ UTRs contain important cis-elements, amongst others, that regulate gene expression. The 3′ UTRs regulate the expression of genes mainly through microRNAs. The regulation of these types of gene expression is also significant under different growth, environmental conditions, and response pathogens, amongst many others.

In this article, the authors first conducted a detailed bioinformatic analysis on 5′ and 3′ UTRs in maize, comparing those to the Arabidopsis and rice gene UTRs. They categorized the UTRs into four gradients (1–500 bp, 501–1000 bp, 1001–2000 bp, and >2000 bp) to study the various elements present in the UTRs.The authors then chose the ZmLAZ1 family and cloned the UTRs to demonstrate how the length of UTR affects gene expression. To elucidate the correlation between 5′ UTR length and gene expression, GUS reporter expression was utilized. Their results indicate that the 5′   UTR of ZmLAZ1 can modulate gene expression and confer functional specificity in a length-dependent manner.

The paper is well-written and has a significant impact on the maize community. The paper should be accepted.

Question: I would like the authors to add a line as to why they chose the ZmLAZ1 Gene Family for further and detailed analysis.

Author Response

Comments 1: The brief report titled “Analysis of 5′ UTR Length-Dependent Control of Gene Expression in Maize: A Case Study with the ZmLAZ1 Gene Family,” analyzed and written by Liu et al is a summary of genome-wide analysis of the 5′ and 3′ UTRs in maize.

The authors have carried out a detailed analysis and have generated a great resource that other researchers can look into for analysis of certain genes and planning further experiments.

UTRs are significant in monitoring gene expression through various mechanisms. The 5′ UTRs contain important cis-elements, amongst others, that regulate gene expression. The 3′ UTRs regulate the expression of genes mainly through microRNAs. The regulation of these types of gene expression is also significant under different growth, environmental conditions, and response pathogens, amongst many others.

In this article, the authors first conducted a detailed bioinformatic analysis on 5′ and 3′ UTRs in maize, comparing those to the Arabidopsis and rice gene UTRs. They categorized the UTRs into four gradients (1–500 bp, 501–1000 bp, 1001–2000 bp, and >2000 bp) to study the various elements present in the UTRs. The authors then chose the ZmLAZ1 family and cloned the UTRs to demonstrate how the length of UTR affects gene expression. To elucidate the correlation between 5′ UTR length and gene expression, GUS reporter expression was utilized. Their results indicate that the 5′ UTR of ZmLAZ1 can modulate gene expression and confer functional specificity in a length-dependent manner.

The paper is well-written and has a significant impact on the maize community. The paper should be accepted.

Response 1: Thank you for your favorable comments and valuable suggestions. We have carefully addressed your concerns in the revised version. We do hope you will be satisfied with our revised version.

Comments 2: I would like the authors to add a line as to why they chose the ZmLAZ1 Gene Family for further and detailed analysis.

Response 2: Thank you very much for your valuable question. According to previous research, the members of the LAZ1 protein family are identified by their conserved DUF300 domains. Despite sharing this domain, their physical and chemical properties, secondary structures, transmembrane structures, transport substrates, subcellular localization, and expression regulation of coding genes are highly diverse. Notably, we found that the 5′ UTR sequence lengths of maize ZmLAZ1 members display a gradient distribution, suggesting a potential relationship between 5′ UTR length and gene function. Therefore, we chose the ZmLAZ1 gene family for further detailed analysis. To clarify the purpose of our study for readers, we have added a description of this section in the Introduction of the revised manuscript.

Reviewer 2 Report

Comments and Suggestions for Authors

I read with interest the manuscript entitled “Analysis of 5′ UTR Length-Dependent Control of Gene Expression in Maize: A Case Study with the ZmLAZ1 Gene Family”. The article is interesting from a scientific and practical point of view and is not objectionable. The subject of the article is important and has great relevance for the scientific environment of the study area. Therefore, the manuscript needs some adjustments so that it can then be forwarded to the publication process. 

- The title accurately reflects the study’s purpose, and the abstract accurately outlines the methods and results. 

- The introduction effectively presents the research problem, and the material and methods section provides detailed information about the studied area.

- The results section is well structured, and illustrated with appropriate graphic material. The results are commented on precisely and clearly. However, in the manuscript, it is noticeable that the discussion of the obtained results in relation to the work of other authors is too poor. In my opinion, the authors should expand this part of the article. This would increase its value.

-  The conclusion part is precise and correspond to the content of the manuscript and the results obtained.

Comments on the Quality of English Language

My suggestion for authors is to enhance the quality of the English language used.

Author Response

Comments 1: I read with interest the manuscript entitled “Analysis of 5′ UTR Length-Dependent Control of Gene Expression in Maize: A Case Study with the ZmLAZ1 Gene Family”. The article is interesting from a scientific and practical point of view and is not objectionable. The subject of the article is important and has great relevance for the scientific environment of the study area. Therefore, the manuscript needs some adjustments so that it can then be forwarded to the publication process.

Response 1: Thank you for your careful reading and valuable comments on our MS. We have tried our best to revise the manuscript accordingly, as suggested by you and other reviewers. Please see detailed descriptions below and in the revised manuscript.

Comments 2: The title accurately reflects the study’s purpose, and the abstract accurately outlines the methods and results.

Response 2: Thank you for your favorable comments.

Comments 3: The introduction effectively presents the research problem, and the material and methods section provides detailed information about the studied area.

Response 3: Thank you for your favorable comments.

Comments 4: The results section is well structured, and illustrated with appropriate graphic material. The results are commented on precisely and clearly. However, in the manuscript, it is noticeable that the discussion of the obtained results in relation to the work of other authors is too poor. In my opinion, the authors should expand this part of the article. This would increase its value.

Response 4: Thank you for pointing this out. In our revised manuscript, we have incorporated the latest research on 5′ UTRs into the Discussion section and expanded on the significance of this study. We also plan to further investigate the function of these ZmLAZ1 gene family members in subsequent studies.  Additionally, we have carefully conducted a thorough linguistic revision to enhance the clarity and readability of our manuscript.

Comments 5: The conclusion part is precise and correspond to the content of the manuscript and the results obtained.

Response 5: Thank you for your favorable comments.